**Data Availability Statement:** The analyzed data in this study is subject to the following licenses/ restrictions: The data contain potentially identifying

# Recurrence of post-term pregnancy and associated factors among women who delivered at Kilimanjaro Christian Medical Centre in northern Tanzania: A retrospective cohort study

**Modesta Mitao**[1]*, **Winfrida C. Mwita**[1], **Cecilia Antony**[1], **Hamidu Adinan**[1], **Benjamin Shayo**[2,3], **Caroline Amour**[1,4], **Innocent B. Mboya**[1,4,5], **Michael Johnson Mahande**[1,2,4]

1 Department of Epidemiology and Biostatistics, Institute of Public Health, Kilimanjaro Christian Medical University College (KCMUCo), Moshi, Tanzania, 2 Department of Obstetrics and Gynaecology, Kilimanjaro Christian Medical Centre, Moshi, Tanzania, 3 Department of Obstetrics and Gynaecology, Baylor College of Medicine, Houston, TX, United States of America, 4 Department of Community Medicine, Institute of Public Health, Kilimanjaro Christian Medical University College (KCMUCo), Moshi, Tanzania, 5 Department of Translational Medicine, Lund University, Malmo, Sweden

* modestamitao@gmail.com

## Abstract

### Background

Post-term pregnancy is a health problem of clinical importance and; tends to recur in subsequent pregnancies. Maternal age, height, and male fetal sex are risk factors associated with Post-term pregnancy. The study aimed to determine the recurrence risk of post-term pregnancy and associated factors among women delivered at KCMC referral hospital.

### Methodology

This retrospective cohort study used KCMC zonal referral hospital medical birth registry cohort data for 43472 women delivered between 2000 and 2018. Data were analyzed using STATA version 15 software. Log-binomial regression with robust variance estimator determined the factors associated recurrence of post-term pregnancy adjusted for other factors.

### Results

A total of 43472 women were analyzed. The proportion of post-term pregnancy was 11.4%, and the recurrence was 14.8%. The recurrence risk of post-term pregnancy was increased when a woman had a history of previous post-term pregnancy (aRR: 1.75; 95%CI: 1.44, 2.11). Advanced maternal age, i.e., ≥35years (aRR: 0.80; 95%CI: 0.65, 0.99), having secondary and higher education (aRR: 0.8; 95%CI: 0.66, 0.97), and being employed (aRR: 0.68; 95%CI: 0.55, 0.84) decreased the recurrence risk of post-term pregnancy. Women with recurrence of post-term pregnancy had a higher risk of delivering newborns weighed ≥4000gm (aRR: 5.05; 95% CI: 2.80, 9.09).

and sensitive patient information. This has also been stipulated by the Local Institutional Review Board of KCMC hospital and the National Ethics Committee in Norway when establishing this birth registry. Permission to use the data in this study was made through the Kilimanjaro Christian Medical University College Research and Ethics Review Committee, and received an approval number PG 03/2020. The authors do not have the legal right to share the data publicly. The Requests to access these datasets should be directed to the Executive Director of the KCMC hospital, kcmcadmin@kcmc.ac.tz.

**Funding:** The author(s) received no specific funding for this work.

**Competing interests:** The authors have declared that no competing interests exist.

**Abbreviations:** ACOG, American College of Obstetricians and Gynecologist; ANC, Antenatal Care; FIGO, Federation of international of Gynecologists and Obstetricians; KCMC, Kilimanjaro Christian Medical Centre; KCMUCo, Kilimanjaro Christian Medical University Collage; MoHCDGEC, Ministry of Health, Community Development, Gender, Elderly and Children; WHO, World Health Organization.

## Conclusion

Post-term pregnancy is associated with recurrence risk in subsequent pregnancies. A history of previous post-term pregnancy is associated risk factor and these women are at increased risk of delivering newborns weighed ≥4000gm. Clinical counselling of women at risk of post-term pregnancy and timely management is recommended to prevent adverse neonatal and maternal outcomes.

## Introduction

Post-term pregnancy extends beyond 42 weeks of gestation (294 days) after the first day of the last menstrual period [1]. Post-term pregnancy is a health problem of clinical importance with recurrence risk in subsequent pregnancies. The global incidence of post-term pregnancy varies by country, the review of randomized clinical trials done in industrialized, low, and middle-income countries reported incidence of between 3% and 14% [2]. In Europe, a study done in 13 countries reported the incidence of post-term pregnancies ranges between 0.5% and 10% [3, 4].

Studies in Sub-Saharan Africa focused on the prevalence of post-term pregnancy. For example, a study in Kenya reported a prevalence of 10% [5], while a study in Ethiopia reported a prevalence of 6% [6].

Previous cohort studies from high-income countries have reported recurrence of post-term pregnancy in subsequent pregnancies ranging from 15% in the Netherland to 16.9% in the USA [7, 8]. There is a paucity of data on the recurrence rate of post-term pregnancies in Sub-Saharan Africa.

Post-term pregnancy is influenced by maternal age, height, male fetal sex, paternal genetics, and behavioral characteristics [8]. Evidence from inter-generational recurrence studies has demonstrated that mothers born post-term have an increased risk of having post-term pregnancy. Similarly, post-term fathers are more likely to trigger post-term pregnancy in their partners [9].

Post-term pregnancy is a health problem associated with an increased risk of maternal, perinatal, fetal, and neonatal morbidity and mortality [3]. Post-term mothers often undergo cesarean delivery and are at increased risk of experiencing postpartum hemorrhage [10–13]. Cesarean section is associated with increased infections, injury to nearby organs, increased need for blood transfusion, death and high associated costs [14].

These women are at increased risk of experiencing shoulder dystocia, which can cause maternal trauma to the bladder, anal sphincter, rectum and arm fracture to the newborn [15].

Intervention such as labor induction at term and beyond term demonstrated a reduction in adverse infant and maternal outcomes such as cesarean section [16]. Also, interventions such as skilled birth attendance coverage, availability of basic and comprehensive emergency obstetric care, promotion of health facility delivery are implemented by the Tanzania Ministry of health to improve maternal and neonatal outcomes including post-term pregnancy [17].

In a systematic review done in Europe, USA, and France reported the risk of perinatal mortality in post-term pregnancy is around 5.8% [4]. Post-term pregnancies recur in subsequent pregnancies, but their magnitude and associated factors are unknown in Tanzania.

Despite the number of interventions in place done by the Tanzania government such as increasing skilled birth attendance coverage, availability of basic and comprehensive emergency obstetric care, promotion of health facility delivery to prevent neonatal and maternal

adverse outcomes caused by obstetric complications such as post-term pregnancy. Therefore, study will help uncover the knowledge gaps and generate data to help design interventions to improve maternal and newborn health and save maternal and children's lives. The study objective was to determine the recurrence risk of post-term pregnancy and associated factors among women delivered at KCMC zonal referral hospital in northern Tanzania.

This study has point out that post-term pregnancy tends to recur in subsequent pregnancies, this knowledge will be used by clinicians to counsel pregnant women at risk for better pregnancy outcome. And policymakers will utilize the study findings to set strategies to reduce the incidence of post-term pregnancy.

## Methods

### Study design and setting

This is a retrospective cohort study utilized birth cohort data from the Kilimanjaro Christian Medical Center (KCMC) zonal referral hospital in Moshi Municipality, Northern Tanzania. The KCMC medical birth registry was established as a pilot in 1999 in collaboration with the Medical Birth Registry of Norway and the University of Bergen.

Data were collected from the department of obstetrics and gynecology as well as the pediatric departments at KCMC hospital. KCMC is one of the four recognized referral hospitals in Tanzania. It serves patients from the Kilimanjaro region (main catchment area), nearby regions, including Arusha, Tanga, Manyara, and other regions of Tanzania. Pregnant women with complications are referred to KCMC for management, while women in the local community may come to deliver by self-referral. The KCMC medical registry has recorded an average of 4,000 births per year. The main socio-economic activities of the Kilimanjaro region include agriculture, tourism and industrial activities [18–20].

### Study population, sample size, and sampling

The parent study included all singleton pregnant women who provided informed consent and recorded in the KCMC medical birth registry between 2000 and 2018. The study excluded participants with missing information on either date of deliveries or date of last menstrual period. There were 62920 deliveries recorded in the KCMC medical birth registry between 200–2018. 62 deliveries with missing hospital numbers were excluded as it was used to link mother's information with their newborns. Also, the following deliveries were excluded 5674 deliveries with gestational age before 28 weeks and after 44 weeks, 3250 multiple births as they are likely to be preterm than post-term from literature, and 2273 deliveries with the missing date of last menstrual period because these are the variables used to compute gestational age which was used to categorize recurrence of post-term pregnancy, the outcome of interest. There were 51490 deliveries from 43742 mothers. We excluded 43472 deliveries without subsequent deliveries in data analysis for recurrence risk of post-term and associated factors. Hence 8025 deliveries with subsequent deliveries were analyzed (**Fig 1**).

### Variables

The main outcome variables in this study were recurrence of post-term pregnancy. Post-term pregnancy is defined as a pregnancy extending beyond 42 weeks of gestation (294 days) from the date of the last menstrual period where by women we were supposed to recall them on their first Antenatal visit [1]. This variable was generated by calculating the difference between the date of the child's birth and the first day of a woman's last menstrual period. Post-term pregnancy was recorded as a binary variable: no post-term pregnancy "<42 weeks of

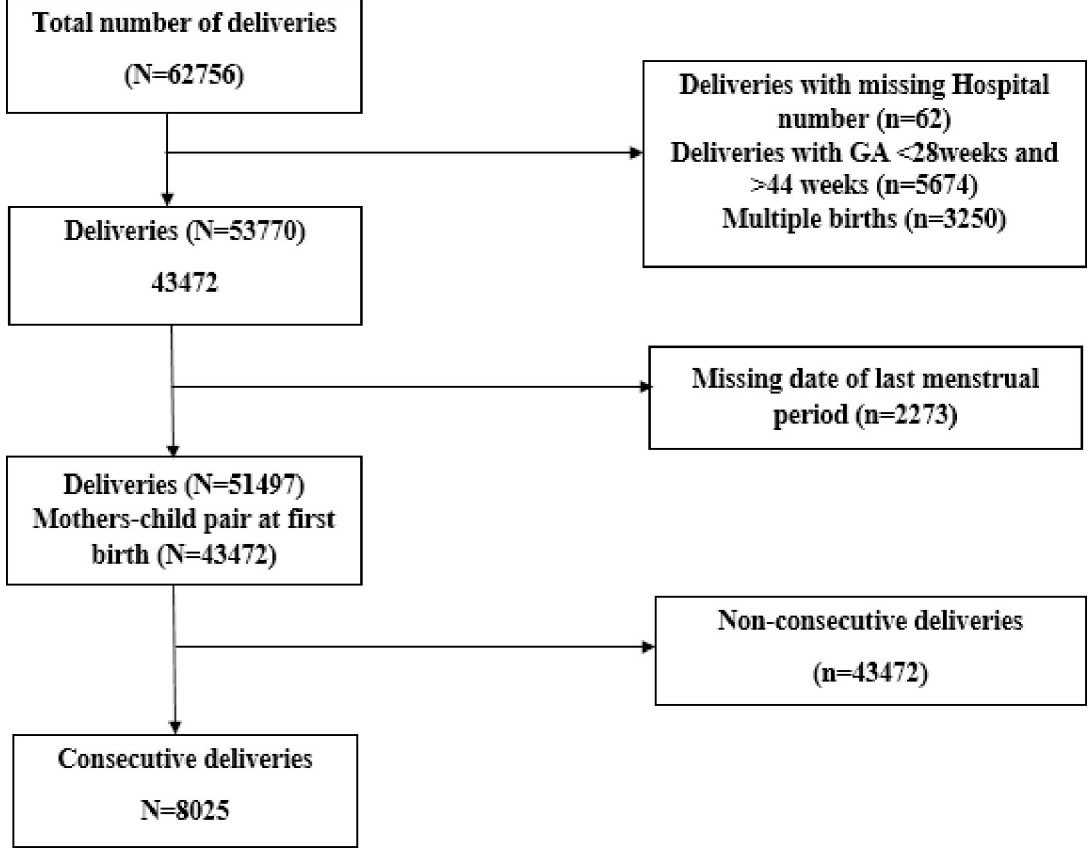

**Fig 1. Flowchart of the number of deliveries from 2000 to 2018 recorded in the KCMC medical birth registry.**

gestational" and post-term pregnancy "≥42 weeks of gestational". Recurrence of post-term pregnancy was defined as the occurrence of one or more post-term pregnancies in a subsequent pregnancy.

Independent variables included maternal socio-demographic characteristics and obstetric characteristics. The study's selection and categorization of independent variables were based on previous literature [18–20].

## Maternal socio-demographic characteristics

Mothers' background characteristics include Mother's age (≤19, 20–34, and ≥35 years), occupation (formal employment, i.e., professions, services and informal employment, i.e., housewife, farmer, students and business), education levels (no formal/primary education and secondary/higher education level), and area of residence (rural and urban). In addition, the study also included height and weight of mother before pregnancy, maternal body mass index (BMI) (underweight ($<18.5$ kg/m$^2$), normal weight (18.5–24.9 kg/m$^2$) and overweight/obesity ($\geq 25$ kg/m$^2$).

## Obstetric characteristic

Obstetric characteristics include birthweight which is defined as a newborn's weight at birth (low birthweight $<2500$gm, normal weight 2500gm- 3999gm, and overweight ≥4000gm) [21];

Child status (Live born, transferred to NICU and perinatal death which is defined as deaths occurring in the first week of life and stillbirths [22]), ANC visit (<4 and ≥ 4 visits).

## Data processing and analysis

Data analysis was performed using STATA version 15 statistical software (Stata Corp, College Station, TX). Descriptive statistics were summarized using the frequency and percentage for categorical variables, and mean with standard deviations was used for numeric variables. The Chi-square test was used to compare the difference in proportions post-term pregnancy by participant characteristics.

The multivariable Log-binomial regression with robust estimator was used to determine the factors associated with recurrence of post-term pregnancy.

All-important variables according to literature were entered in final models to estimate its contribution on recurrence of post-term pregnancy, the outcome of interest. A variable was a confounder if its inclusion in the model changed the relative risk by 10%. Also, multicollinearity between exposures were checked using pairwise correlation. A p-value of <5% was considered statistically significant.

## Ethical consideration

Ethical clearance for the KCMC Medical birth registry was granted from the ethics committee at KCMC and the National Ethics of Norway. Also, similar clearance has been granted from the Tanzanian Ministry of health, Commission for Science and Technology. Verbal consent was sought from each mother prior to the interview, which was conducted just after the woman had given birth. Participation was voluntary and had no impact on the management women would receive. Mothers were free to refuse to reply to any single question. For privacy and confidentiality, unique identification numbers were used to both identify and then link mothers with child records.

Ethical approval to carry out the current study was obtained from the Kilimanjaro Christian Medical College Research Ethics and Review Committee (KCMU-CREC) with clearance number PG 03/2020.

## Results

### Socio-demographic and obstetric characteristics of women in a subsequent pregnancy

A total of 8025 deliveries were analyzed. The recurrence risk of post-term pregnancy in subsequent pregnancies (two or more pregnancies) was 14.8% (143/965). The proportion of post-term pregnancy varies across socio-demographic and obstetrics characteristics in the subsequent pregnancies (**Table 1**). The distribution of preterm, term, and post-term births by year shown (**Fig 2**). 10.4% of women who had no formal and primary education had a recurrence of post-term pregnancy compared to 7.5% of those who had secondary and higher education; P-value <0.001. 10% of unemployed women had recurrence of post-term pregnancy compared to 6.1% of employed women; P-value<0.001. 13.5% of women who gave birth to children weighing 4000gm or more had recurrence of post-term pregnancy; P-value<0.001.

### Factors associated with recurrence of post-term pregnancy

The factors associated with the recurrence of post-term pregnancy are displayed in **Table 2**. In crude analysis, previous post-term pregnancy (cRR: 1.81; 95%CI: 1.49, 2.19) was significantly associated with an increased risk of recurrent postterm pregnancy in the subsequent

**Table 1. Socio-demographic and obstetric characteristics of women delivered at KCMC in subsequent pregnancy, 2000–2018 (N = 8025).**

| Characteristics | Total (n) | Post-term | P-value |
|---|---|---|---|
| **Previous post-term** | | | |
| No | 7060 | 580 (8.2) | <0.001 |
| Yes | 965 | 143 (14.8) | |
| **Mother's Age (years)[a]** | | | |
| 15–19 | 111 | 14(12.6) | 0.056 |
| 20–24 | 1112 | 116(10.4) | |
| 25–34 | 5075 | 458 (9.0)) | |
| ≥35 | 1706 | 133 (7.8) | |
| **Residence [b]** | | | |
| Rural | 2573 | 238 (9.3) | 0.615 |
| Urban | 5446 | 485 (8.9) | |
| **Mother's education level [c]** | | | |
| No formal and primary education | 4087 | 426 (10.4) | <0.001 |
| Secondary and Higher | 3930 | 294 (7.5) | |
| **Mother's occupation [d]** | | | |
| Unemployed | 5963 | 596 (10.0) | <0.001 |
| Employed | 2037 | 125 (6.1) | |
| **BMI[e]** | | | |
| Underweight | 466 | 36 (7.7) | 0.394 |
| Normal weight | 3543 | 321 (9.1) | |
| Overweight/obesity | 2183 | 211 (9.7) | |
| **Mode of delivery[f]** | | | |
| Spontaneous delivery | 5271 | 512 (9.71) | 0.008 |
| Assisted delivery | 122 | 11 (9.02) | |
| Cesarean section | 2601 | 197 (7.57) | |
| **Induction of labour[g]** | | | |
| No | 5761 | 520 (9.03) | 0.673 |
| Yes | 2210 | 194 (8.78) | |
| **ANC visit[h]** | | | |
| <4 | 2680 | 209 (7.8) | 0.006 |
| ≥4 | 5273 | 510 (9.7) | |
| **Child characteristics** | | | |
| **Child status [i]** | | | |
| Live born | 6833 | 620 (9.1) | 0.172 |
| Transferred to NICU | 995 | 92 (9.2) | |
| Perinatal death | 163 | 78 (4.29) | |
| **Birth weight(g) [j]** | | | |
| <2500 | 640 | 22 (3.4) | <0.001 |
| 2500–3999 | 6926 | 642 (9.3) | |
| ≥4000 | 437 | 59 (13.5) | |
| **Year of birth** | | | |
| 2000–2010 | 3871 | 363(9.4) | 0.994 |
| 2011–2018 | 4154 | 363(8.7) | |

a: n = 8004; b: n = 8019; c: n = 7971; d: n = 8000; e: n = 6192; f: n = 7481; g: n = 7971; h: n = 7953; i: n = 7991; j: n = 8003

*Frequencies (n) do not tally to the total due to missing values in these variables

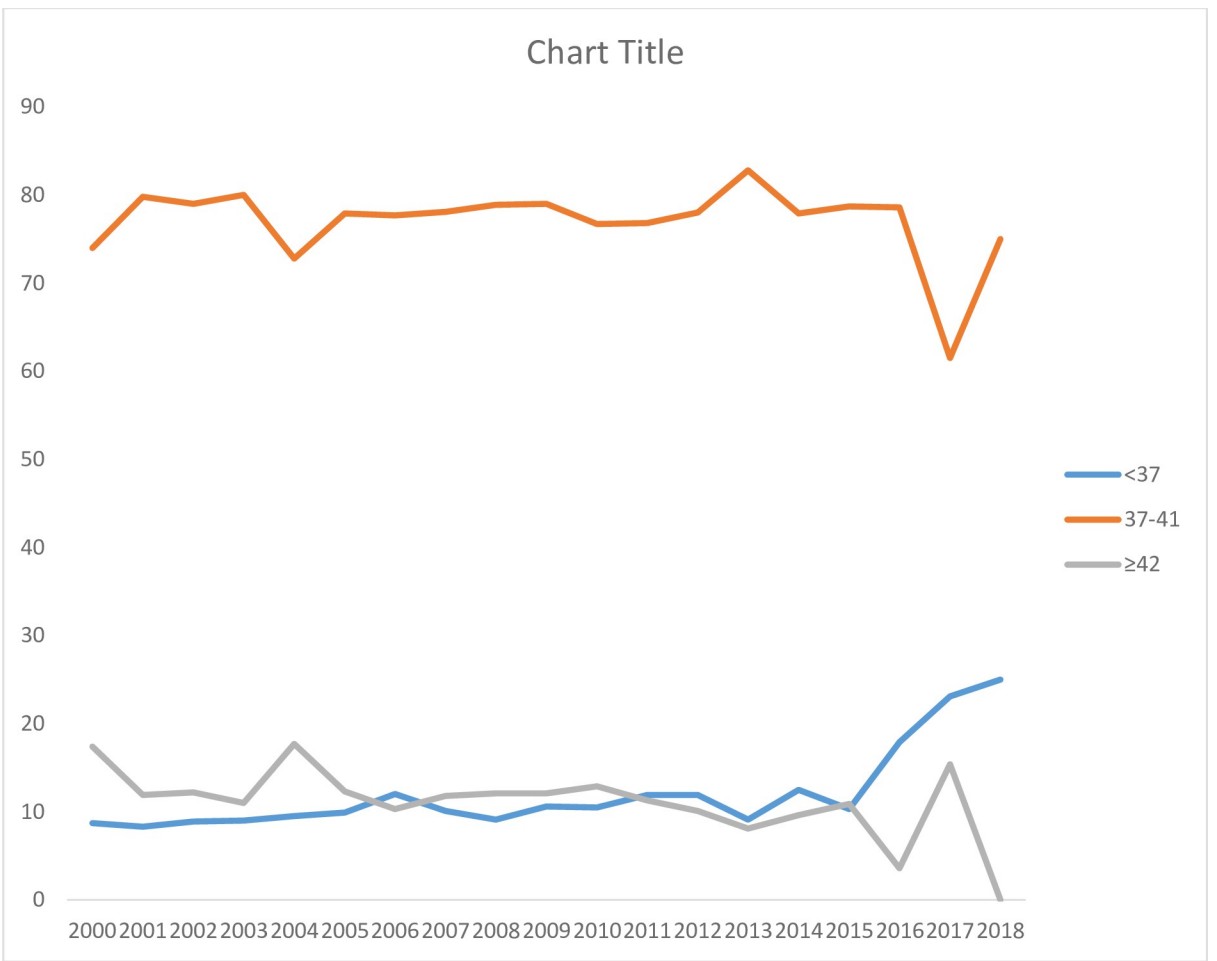

**Fig 2. Distribution of pre-term, term, and post-term by year among women delivered at KCMC in 2000–2018 (N = 8025).**

pregnancies. Other factors such as young maternal age (15–19) years (cRR: 1.21; 95%CI: 0.70, 2.08), 20–24 years (cRR: 1.22; 95%CI: 0.98, 1.52) increased women's likelihood of recurrence post-term pregnancy but these were not statistically significant. On the other hand, secondary and higher education (cRR: 0.78; 95%CI: 0.65, 0.95),employed women (cRR: 0.73; 95% CI: 0.57, 0.94) and having cesarean section delivery were significantly associated with a lower risk of recurrent post-term pregnancy compared to counterparts.

In adjusted analysis, the previous history of post-term pregnancy remained significantly associated with an increased risk of post-term pregnancy (aRR: 1.81; 95%CI: 1.49, 2.19). Advanced maternal age, i.e., ≥35years (aRR: 0.78; 95%CI: 0.63, 0.96), and employed women (aRR: 0.73; 95%CI: 0.57, 0.94) had lower risk of recurrence of post-term pregnancy compared to their counterparts. Women who had recurrent post-term pregnancies had a 5-fold higher risk of delivering heavier newborns (aRR: 4.99; 95%CI: 2.71, 9.19).

## Discussion

This study aimed to determine the recurrence risk of post-term pregnancy and associated factors among women who delivered at KCMC from 2000 to 2018. The proportion of women who had post-term pregnancies during their first recorded pregnancy was 11.4%. The

**Table 2. Factors associated with recurrence of post-term pregnancy among women delivered at KCMC in subsequent pregnancy, 2000–2018 (N = 8025).**

| Characteristics | cRR (95%CI) | P-value | aRR (95%CI) | P-value |
|---|---|---|---|---|
| **Previous post-term** | | | | |
| No | 1.00 | | 1.00 | |
| Yes | 1.74 (1.52–2.14) | <0.001 | 1.81 (1.49–2.19) | <0.001 |
| **Mother's Age(years)** | | | | |
| 25–34 | 1.00 | | 1.00 | |
| 15–19 | 1.21(0.70–2.08) | 0.491 | 1.40(0.76–2.59) | 0.280 |
| 20–24 | 1.15 (0.95–1.39) | 0.153 | 1.22(0.98–1.52) | 0.080 |
| ≥35 | 0.86 (0.71–1.03) | 0.101 | 0.78 (0.63–0.96) | 0.020 |
| **Mother's education level** | | | | |
| No formal and Primary education | 1.00 | | 1.00 | |
| Secondary and Higher | 0.72 (0.63–0.83) | <0.001 | 0.78(0.65–0.95) | 0.012 |
| **Mother's occupation** | | | | |
| Unemployed | 1.00 | | 1.00 | |
| Employed | 0.61 (0.51–0.74) | <0.001 | 0.73 (0.57–0.94) | 0.013 |
| **ANC visit** | | | | |
| <4 | 1.00 | | 1.00 | |
| ≥4 | 1.26(1.08–1.48) | 0.003 | 1.29 (1.08–1.55) | 0.005 |
| **Mode of delivery** | | | | |
| Spontaneous delivery | 1.00 | | 1.00 | |
| Assisted delivery | 1.01(0.58–1.73) | 0.979 | 1.02(0.58–1.81) | 0.921 |
| Cesarean section | 0.78(0.66–0.91) | 0.002 | 0.73(0.60–0.88) | 0.001 |
| **Induction of labour** | | | | |
| No | 1.00 | | 1.00 | |
| Yes | 0.95(0.81–1.12) | 0.550 | 0.89(0.73–1.06) | 0.185 |
| **BMI** | | | | |
| Normal weight | 1.00 | | 1.00 | |
| Underweight | 0.85 (0.61–1.19) | 0.345 | 1.22(0.83–1.52) | 0.348 |
| Overweight/Obese | 1.07 (0.90–1.26) | 0.443 | 1.18 (0.62–2.24) | 0.108 |
| **Child status** | | | | |
| Live | 1.00 | | 1.00 | |
| Transferred to NICU | 1.01 (0.84–1.27) | 0.752 | 1.12 (0.80–1.36) | 0.759 |
| Perinatal death | 0.60(0.3–1.13) | 0.114 | 1.11 (0.45–1.64) | 0.747 |
| **Birth weight(g)** | | | | |
| <2500 | 1.00 | | 1.00 | |
| 2500–3999 | 2.70 (1.78–4.09) | <0.001 | 3.24 (1.83–5.74) | <0.001 |
| ≥4000 | 4.02 (2.51–6.44) | <0.001 | 4.99 (2.71–9.19) | <0.001 |
| **Year of birth** | | | | |
| 1st decade (2000–2010) | 1.00 | 0.319 | 1.00 | 0.560 |
| 2nd decade (2011–2018) | 0.93(0.81–1.07) | | 1.05(0.89–1.25) | |

* cRR- Crude relative risk

* aRR-Adjusted relative risk

*Adjusted for previous post-term, mother's age, education, occupation, ANC visit, BMI, mode of delivery, induction of labour, child status, and child's birthweight.

recurrence risk of post-term pregnancy was 14.8%. In addition, a previous history of post-term pregnancy was associated with an increased risk of post-term pregnancy. In contrast, advanced maternal age, being employed were significantly associated with a lower risk of recurrence of

post-term pregnancy. Furthermore, recurrent post-term pregnancy increased women's likelihood of delivering heavier newborns.

The recurrence risk of post-term pregnancy in the study setting was 14.8%. This figure is approximate similar to 15% which was previously reported in the Netherland [8]. But it is lower compared to 16.9% and 19.9% which were reported in the USA and Denmark [7, 23]. Difference in management guidelines of post-term pregnancy across countries may also explain variations in the recurrence risk, in countries which have adopted WHO recommendation of induction of labor at 41 weeks would have low rate of recurrence of post-term pregnancy [16].

In this study, advanced maternal age (>35 years) was associated with lower risk of recurrent post-term pregnancy than women aged 25–34 years. Similarly, women with secondary and higher education had a lower risk of recurrence of post-term pregnancy. This is consistent with the previous study in the USA [7]. It is possible that higher education influences awareness of health education. Therefore, educated women were more likely to seek health care by attending health facilities early when the due date had passed when compared with their counterparts with lower education levels [24].

In this study, cesarean section delivery was associated with lower risk of recurrent post-term pregnancy. In contrast, studies in SSA have shown that recurrence risk of adverse pregnancy outcomes such as shoulder dystocia and fetal macrosomia [13, 25, 26]. Similar factors are also common in Tanzania. These factors could also be associated with an increased recurrence risk of post-term pregnancy.

The previous history of previous post-term pregnancy was associated with an increased risk of recurrent post-term pregnancy. Our finding is in-congruent with previous studies in the Netherlands, Denmark, Scotland, and the USA [7, 8, 23, 27, 28]. This could be because post-term pregnancy is influenced by genetic factors responsible for controlling parturition, which can persist between successive pregnancies [7].

The study strength is that it is the first in Tanzania to evaluate the recurrence risk and factors associated with post-term pregnancy. Therefore, the study estimated the burden of recurrent post-term pregnancy and associated factors in northern Tanzania. The KCMC Medical birth registry data used in this study has large sample size, hence enough statistical power that increase precision of the estimates. In addition, the cohort design nature using maternally linked data to create reproductive history for each woman enabled estimating the recurrence risk of post-term pregnancy and its determinants.

Despite these strengths, it is worth noting that this study utilized secondary data, which might have compromised the data quality and thereby the estimates. This includes missing values in outcome and exposures. Some reproductive parameters to study recurrent post-term depend on maternal recall memories, subject to recall bias. For example, mothers were asked to recall the date of their last normal menstrual period at the time. Since the majority of the women can't exactly remember the date of their last menstrual period, this can lead to over or underestimating gestational age and consequently post-term pregnancy. Also, if this happened, it could lead to differential misclassification bias. We acknowledge the missing of ultrasound scans at the first trimester, which is most accurate means of estimating gestation age.

In addition, since our study utilized data from the referral setting, there is a possibility of selection bias. However, more than two-thirds of our study participants were self-referred.

## Conclusion

The proportion of post-term pregnancies in this study was 11.4%, while the recurrence risk of post-term pregnancy 14.8%. Recurrence of post-term pregnancy was associated with having

history of previous post-term pregnancy, advanced maternal age ≥35years, being employed, and delivering newborns weighed ≥4000gm.

## Recommendations

The study recommends health workers to provide health education to women on risk factors associated with recurrence of post-term pregnancy such history of post-term pregnancy. Hence, women at risk should attend to health facility early for proper and timely management for best pregnancy outcomes.

Women should be encouraged by health workers to attend at health facility when the due date has been reached without initiation of labour for interventions such as labour induction to prevent adverse perinatal and maternal outcomes.

## Acknowledgments

We acknowledge all nurses and doctors at KCMC who took part in data collection and all women who consented to provide their information making this study possible. Also, we would like to acknowledge Dr. Joachim Frank Magoma, Dr. Rafiki Mjema, and Dr. Joel Msafiri Francis who reviewed this work for their valuable contribution.

## Author Contributions

**Conceptualization:** Innocent B. Mboya, Michael Johnson Mahande.

**Data curation:** Innocent B. Mboya.

**Formal analysis:** Modesta Mitao, Winfrida C. Mwita, Cecilia Antony, Michael Johnson Mahande.

**Methodology:** Modesta Mitao, Winfrida C. Mwita, Cecilia Antony, Hamidu Adinan, Caroline Amour, Innocent B. Mboya, Michael Johnson Mahande.

**Supervision:** Caroline Amour, Innocent B. Mboya, Michael Johnson Mahande.

**Validation:** Innocent B. Mboya.

**Writing – original draft:** Modesta Mitao, Innocent B. Mboya, Michael Johnson Mahande.

**Writing – review & editing:** Modesta Mitao, Benjamin Shayo, Caroline Amour, Innocent B. Mboya, Michael Johnson Mahande.

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
