## [Decision Letter · Decision Letter 0]

22 Mar 2022

PONE-D-21-33495Recurrence of post-term pregnancy and associated factors among women who delivered at Kilimanjaro Christian Medical University Centre in Northern Tanzania: A retrospective cohort studyPLOS ONE

Dear Dr. Mitao,

Thank you for submitting your manuscript to PLOS ONE. After careful consideration, we feel that it has merit but does not fully meet PLOS ONE’s publication criteria as it currently stands. Therefore, we invite you to submit a revised version of the manuscript that addresses the points raised during the review process.

Interesting findings, but there are many rooms to develop the current version. Firstly introduction section needs to be revised as structured an introduction with sharp objectives and rationale of analyzing this data without considering time (2 decades old data). Method section lacks of defining data, eg., why parity is missing, why and how data were merged to a converge model instead of considering poison model. Needs appropriate reflection of results, discussion section is currently lacking with inference of findings and support of relevant literature and epilogue.  Please submit your revised manuscript by May 06 2022 11:59PM. If you will need more time than this to complete your revisions, please reply to this message or contact the journal office at plosone@plos.org. Please include the following items when submitting your revised manuscript:A rebuttal letter that responds to each point raised by the academic editor and reviewer(s). You should upload this letter as a separate file labeled 'Response to Reviewers'.A marked-up copy of your manuscript that highlights changes made to the original version. You should upload this as a separate file labeled 'Revised Manuscript with Track Changes'.An unmarked version of your revised paper without tracked changes. You should upload this as a separate file labeled 'Manuscript'.

We look forward to receiving your revised manuscript.

Kind regards,

Mahfuzar Rahman, MD, PhD

Academic Editor

PLOS ONE

Journal Requirements:

3. Please include your tables as part of your main manuscript and remove the individual files. Please note that supplementary tables (should remain/ be uploaded) as separate "supporting information" files.

Reviewers' comments:

Reviewer's Responses to Questions

**Comments to the Author**

1. Is the manuscript technically sound, and do the data support the conclusions?

Reviewer #1: Yes

Reviewer #2: No

2. Has the statistical analysis been performed appropriately and rigorously? 

Reviewer #1: Yes

Reviewer #2: No

3. Have the authors made all data underlying the findings in their manuscript fully available?

Reviewer #1: No

Reviewer #2: No

4. Is the manuscript presented in an intelligible fashion and written in standard English?

Reviewer #1: Yes

Reviewer #2: No

5. Review Comments to the Author

Reviewer #1: Abstract:

Data analysis: I am wondering why to use robust variance estimator in log-binomial model if the model is converged. Often multivariable log-binomial model fails to converge and, in that case, we use Poisson model with robust variance estimator. Please clarify what exactly was done.

Line 94: Please write as “This is a retrospective study….”

Line 111: Please write “2000-2018”

Line 113: Please check with the figure-1. Here it is mentioned after 44 weeks whereas in the figure it is mentioned >45 weeks, which are not same.

Line 150/151: Please see the comments in the abstract.

Line 152: Please justify using P value of 5% in bivariate analysis for considering for multivariable model.

Some minor English language editing is needed.

Reviewer #2: The authors evaluated the proportion of post-term birth and its associations with selected socio-demographic, service utilization, and nutritional factors. I suggest the authors address the points below to improve the clarity and maximize the use of available data and discuss in a more focused way based on the study results.

1) The authors should describe the distribution of preterm, term, and post-term births by year to get general ideas about the burden of those events in the hospital.

2) The data collected covered the time of almost two decades. A significant change in social and service utilization may occur during that period. The authors did not consider the time in the analysis. There is also a change in cesarean section (CS) rates, and the history of CS may influence the post-term rate in subsequent pregnancies.

3) The authors should consider revising the abstract. In the background section, the sentence looks like a conclusion (P 2, L 28-30). As mentioned in the results, 'the risk of 14.8%' should be a proportion. In the end, the conclusion is not based on study results and therefore needs revision.

4) Introduction section should be more organized. For example, the effect of post-term birth has been mentioned in several paragraphs. It is not clear how the present study will help clinicians and policymakers (P 3, L 64,65)? There is a lack of texts to indicate the weaknesses of the previous studies. Furthermore, how the present study will contribute to our understanding of the problem. Why is there a lack of studies in low-income countries? Please revise the sentence (P 3, L 69-71). The last sentence is not meaning any sense. The study's rationale should be modified to convince the study's usefulness.

5) The methods section should describe how each variable was measured. Please add texts to explain how women's LMP, height, and weight were collected? There is no mention of parity. How many women are with their first and second births available? It may be interesting to see the recurrence risk in the second birth.

6) How the missing information was dealt with in the analysis. For example, table 2 shows a significant number of missing values. The authors should clarify how many women were included in the adjusted analysis.

7) Discussion section: This section needs more work to make the article interesting. The essential findings are not highlighted adequately. It is implied that the induction of labor or delivery by cesarean section may influence the results. What are the implications of this unavailability of data? What was the recall period of LMP date information? It is also not convincing how education may influence the occurrence of post-term births. Furthermore, it need more intuitive discussion how the missing values may bias the study findings?

8) The authors also should check the language.

6. PLOS authors have the option to publish the peer review history of their article (what does this mean?). If published, this will include your full peer review and any attached files.

Reviewer #1: No

Reviewer #2: **Yes: **Anisur Rahman

---

## [Author Response · Author response to Decision Letter 0]

20 Aug 2022

The authors should consider revising the abstract. In the background section, the sentence looks like a conclusion (P 2, L 28-30). As mentioned in the results, 'the risk of 14.8%' should be a proportion. In the end, the conclusion is not based on study results and therefore needs revision.

Response:The abstract in background section has been revised, the concluding sentence have been removed (P2, L28-29). The conclusion section has been revised to base on study results.

---

## [Decision Letter · Decision Letter 1]

8 Feb 2023

Recurrence of post-term pregnancy and associated factors among women who delivered at Kilimanjaro Christian Medical University Centre in Northern Tanzania: A retrospective cohort study

PONE-D-21-33495R1

Dear Dr. Mitao,

We’re pleased to inform you that your manuscript has been judged scientifically suitable for publication and will be formally accepted for publication once it meets all outstanding technical requirements.

Kind regards,

Mahfuzar Rahman, MD, PhD

Academic Editor

PLOS ONE

Additional Editor Comments (optional):

Reviewers' comments:

Reviewer's Responses to Questions

**Comments to the Author**

1. If the authors have adequately addressed your comments raised in a previous round of review and you feel that this manuscript is now acceptable for publication, you may indicate that here to bypass the “Comments to the Author” section, enter your conflict of interest statement in the “Confidential to Editor” section, and submit your "Accept" recommendation.

Reviewer #1: All comments have been addressed

2. Is the manuscript technically sound, and do the data support the conclusions?

Reviewer #1: Yes

3. Has the statistical analysis been performed appropriately and rigorously? 

Reviewer #1: Yes

4. Have the authors made all data underlying the findings in their manuscript fully available?

Reviewer #1: No

5. Is the manuscript presented in an intelligible fashion and written in standard English?

Reviewer #1: Yes

6. Review Comments to the Author

Reviewer #1: As the authors addressed all the issues raised earlier satisfactorily, now this manuscript can be published.

7. PLOS authors have the option to publish the peer review history of their article (what does this mean?). If published, this will include your full peer review and any attached files.

Reviewer #1: **Yes: **Dipak Kumar Mitra

---

## [Editor Report · Acceptance letter]

14 Feb 2023

PONE-D-21-33495R1 

Recurrence of post-term pregnancy and associated factors among women who delivered at Kilimanjaro Christian Medical Centre in northern Tanzania: A retrospective cohort study 

Dear Dr. Mitao:

I'm pleased to inform you that your manuscript has been deemed suitable for publication in PLOS ONE. Congratulations! Your manuscript is now with our production department. 

Kind regards, 

on behalf of

Dr. Mahfuzar Rahman 

Academic Editor

PLOS ONE